# Influence of UAS Flight Altitude and Speed on Aboveground Biomass Prediction

Neal C. Swayze [1,*], Wade T. Tinkham [2], Matthew B. Creasy [2], Jody C. Vogeler [1], Chad M. Hoffman [2] and Andrew T. Hudak [3]

[1] Natural Resource Ecology Laboratory, Colorado State University, Fort Collins, CO 80523, USA; jody.vogeler@colostate.edu

[2] Department of Forest and Rangeland Stewardship, Colorado State University, Fort Collins, CO 80524, USA; wade.tinkham@colostate.edu (W.T.T.); matthew.creasy@colostate.edu (M.B.C.); c.hoffman@colostate.edu (C.M.H.)

[3] Rocky Mountain Research Station, United States Department of Agriculture Forest Service, Moscow, ID 83844, USA; andrew.hudak@usda.gov

\* Correspondence: neal.swayze@colostate.edu

**Abstract:** The management of low-density savannah and woodland forests for carbon storage presents a mechanism to offset the expense of ecologically informed forest management strategies. However, existing carbon monitoring systems draw on vast amounts of either field observations or aerial light detection and ranging (LiDAR) collections, making them financially prohibitive in low productivity systems where forest management focuses on promoting resilience to disturbance and multiple uses. This study evaluates how UAS altitude and flight speed influence area-based aboveground forest biomass model predictions. The imagery was acquired across a range of UAS altitudes and flight speeds that influence the efficiency of data collection. Data were processed using common structures from motion photogrammetry algorithms and then modeled using Random Forest. These results are compared to LiDAR observations collected from fixed-wing manned aircraft and modeled using the same routine. Results show a strong positive relationship between flight altitude and plot-based aboveground biomass modeling accuracy. UAS predictions increasingly outperformed (2–24% increased variance explained) commercial airborne LiDAR strategies as acquisition altitude increased from 80–120 m. The reduced cost of UAS data collection and processing and improved biomass modeling accuracy over airborne LiDAR approaches could make carbon monitoring viable in low productivity forest systems.

**Keywords:** structure from motion; carbon; monitoring; area-based; random forest; uav; forest; woodland

## 1. Introduction

Savannah, woodland, and dry forest types with low tree densities occupy vast regions of every continent [1] and provide a mixture of land uses, including livestock grazing, wildlife habitat, agriculture, and fuelwood production. Carbon management has been proposed in these regions as an alternative land-use strategy that would promote maintenance of existing tree cover and prevent or reverse land conversion through deforestation [2]. Existing carbon monitoring system protocols rely on significant field observations or aerial light detection and ranging (LiDAR) collections [3]. Although such data collection strategies are widely used in productive temperate and tropical forest systems, they can be financially prohibitive within low productivity systems where forest management focuses on system resilience to disturbance and promoting multiple uses instead of the limited timber value [4,5]. The little commercial timber harvesting within these regions has restricted aerial LiDAR availability. Existing datasets are often collected for terrain mapping at pulse densities (0.5–2 pulses m$^{-2}$), which are not ideal for characterizing vegetation [6]. Developing alternative, cost-effective aboveground biomass monitoring strategies could

provide carbon market funding opportunities to prevent the land conversion of these valuable ecosystems.

Although aerial LiDAR has been widely shown to produce reliable area-based and individual tree estimates of forest density (i.e., basal area per hectare [7]) and biomass [8–10], the availability of LiDAR of sufficient point density in these savannahs and woodlands has limited its usefulness for vegetation applications. Unmanned Aerial Systems (UAS) have emerged as a localized monitoring alternative providing spatially continuous observations at high spatial resolution and potentially more frequent temporal intervals. Professional grade UAS have recently become more accessible to consumers with entry-level equipment costs under $2000. Despite the low price point, UAS typically has high-accuracy GPS receivers, automated inertial navigation systems, and sensors capable of very high-resolution (VHR; <10 cm) optical and active remote sensing [11]. This combination of technologies is ideal for photogrammetry, which requires high-resolution, spatially accurate images.

The high-cost and temporal resolution limitations of aerial LiDAR are where UAS can fill data needs for forest management organizations; for monitoring at management relevant scales, UAS-based approaches can cost orders of magnitude less than aircraft-based LiDAR acquisitions and can be flown as often as favorable conditions are met. Over the last decade, numerous studies have utilized UAS to generate VHR two-dimensional orthomosaic maps [12,13] and three-dimensional point clouds of forest structure and terrain using various structure from motion (SfM) processing algorithms [14,15]. SfM algorithms identify common points within overlapping images and, through a geometric process utilizing the position and rotation of captured images, a three-dimensional point cloud is generated [15]. Due to the high degree of image overlap, SfM point clouds can have data densities exceeding 1000 points $m^{-2}$ compared to LiDAR's common 4–30 points $m^{-2}$, and therefore can provide a more detailed representation of fine-scale forest structure compared to LiDAR. Linking the spectral data from UAS imagery to SfM point clouds during forest biomass modeling can improve UAS prediction accuracy by up to 80% over standard point cloud-only models [16].

In pursuit of operationalizing UAS technology for forest monitoring, several studies have evaluated the influence of acquisition parameters on data quality and forest structure characterization accuracy. Dandois et al. [17] demonstrated that increasing the forward and side overlap levels up to 80% led to improved location and height accuracies in forested environments. When controlling forward and side overlap separately, Seifert et al. [18] found that maintaining high (>90%) forward overlap with lower side overlap (~70%) provided a balance between data accuracy, flight time, area coverage, and data processing time. These studies suggest that forward overlap should be maximized as it has minimal impact on flight time, or the area covered in a single acquisition. In contrast, the level of side overlap should balance being significant enough for image alignment without sacrificing acquisition extent. However, other acquisition parameters have provided unclear results. Fraser and Congalton [13] found that flying at 100 m above the tree canopy provided the best image alignment, while Torres-Sánchez et al. [12] and Swayze et al. [19] found no significant impact of altitude on object-based canopy parameter extraction. At the same time, Seifert et al. [18] found that flights within 15 to 20 m of the vegetation canopy (or 2 to 2.33 times maximum tree heights) resulted in significantly more image registration points with improved precision. While these results indicate a range of optimal parameters for UAS image alignment within different vegetation types, significant knowledge gaps exist in how acquisition parameters will impact UAS-based products derived from the entire point cloud instead of orthophotographs or canopy height models. Clarity is needed on how acquisition parameters ultimately impact observations of forest structures and derived estimates such as aboveground biomass.

This study examines how flight altitude and speed impact UAS area-based predictions of forest biomass compared to standard aerial LiDAR modeling strategies across a range of forest structures found in ponderosa pine (*Pinus ponderosa* var. *scopulorum* Dougl. Ex Laws.) dominated woodlands and forests. Specifically, variance explained and precision

of UAS-based models of forest biomass is standardized against a LiDAR-based model to examine how flight altitude and speed impact model reliability. Additionally, this analysis will investigate how segmentation of SfM point clouds based on spectral indices impacts model performance.

## 2. Materials and Methods

### 2.1. Study Area and Field Data

The study occurred at two ponderosa pine forests with existing aerial LiDAR and stem-mapped forest inventories in the central Rocky Mountains (Figure 1). A total of five 60 m × 100 m study units were selected to represent a range of forest densities. The Lookout Canyon site is in the Kaibab National Forest in Northern Arizona, ~65 km southeast of Kanab, Utah, at 2400 m elevation with slopes <10%. The forest is dominated by ponderosa pine and was divided into three 4-hectare stands for thinning, including a control stand and two stands thinned to 9.2 and 13.8 m$^2$ ha$^{-1}$ of basal area in 1993 (Table 1), hereafter KNF1, KNF2, and KNF3. Following thinning, quaking aspen (*Populus tremuloides Michx.*) began reestablishing in the understory alongside cycles of ponderosa pine regeneration. A prescribed fire in an adjacent stand escaped in 1999 and burnt through the understory, killing more than 600 small-diameter trees (~15% of stem density).

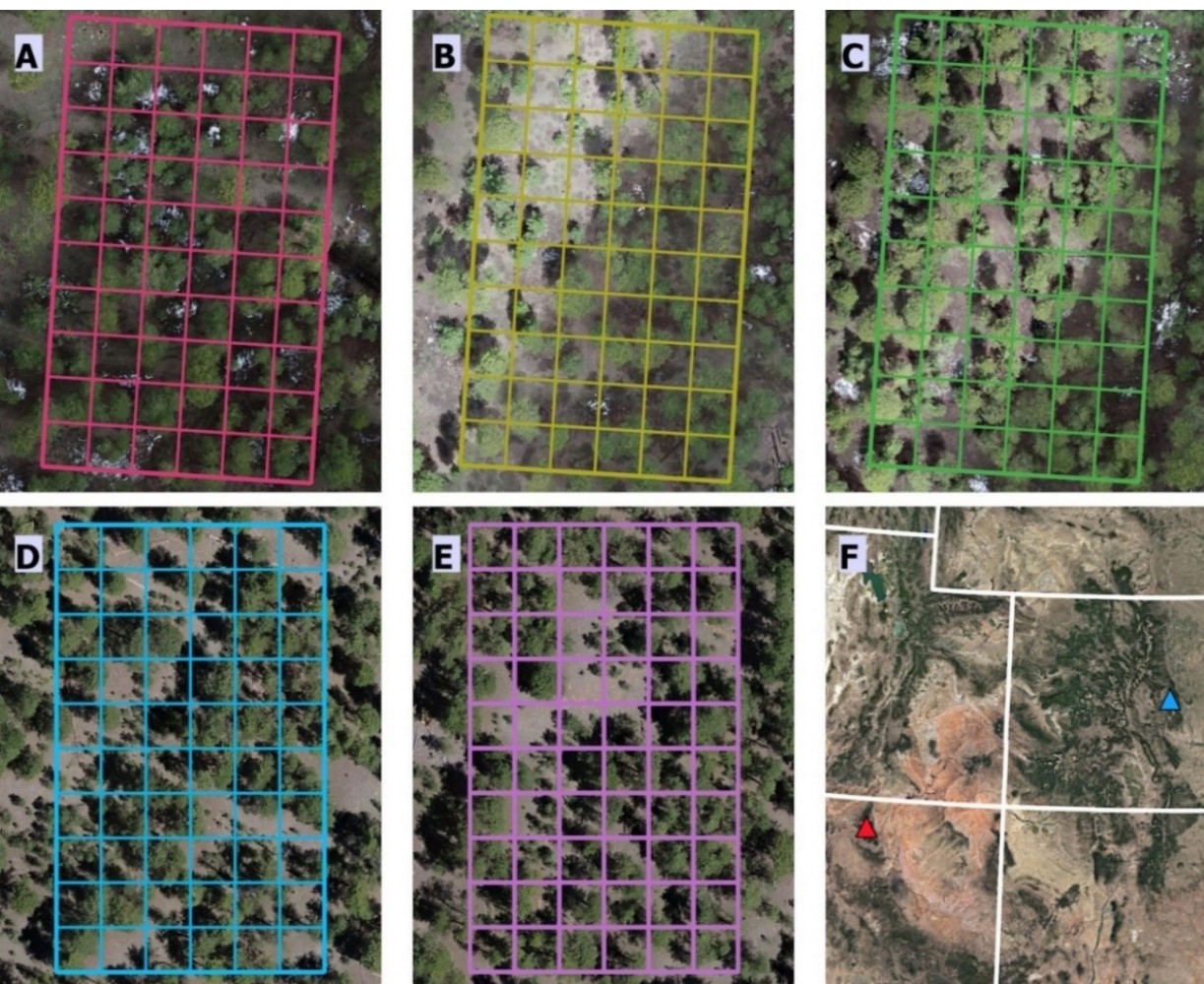

**Figure 1.** Five 60 m × 100 m study areas at the Kaibab National Forest in Northern Arizona (KNF1: (**A**), KNF2: (**B**), KNF3: (**C**)) and Manitou Experimental Forest in Central Colorado (MEF1: (**D**), MEF2: (**E**)), with the location of KNF study area (red) and MEF study area (blue) displayed in panel (**F**). Each study area was divided into sixty 10 m × 10 m plots.

The N1 forest dynamics site is located within the Manitou Experimental Forest on the Pike-San Isabel National Forest, 40 km northwest of Colorado Springs, Colorado. The average elevation is 2500 m, with a mild slope (<5%). This site is a multi-aged montane ponderosa pine forest that was selective logged between 1880 and 1886 [20]. After logging, the forest was undisturbed with no documented fires since 1846 and only minor mountain pine beetle disturbance in the late 1970s. In the 140 years following harvest, several regeneration pulses have led to varying forest densities, with minor components of Douglas-fir (*Pseudotsuga menziesii* (Mirb.) *Franco* var. *glauca* (Beissn.) Franco) and blue spruce (*Picea pungens Engelm.*) in the understory. Native grasses and a few low-growing woody shrubs occur in the understory. Two 60 m × 100 m study units were established at N1 in areas with different levels of ingrowth, hereafter referred to as MEF1 and MEF2 (Table 1).

**Table 1.** Location and forest stand structure at the Kaibab National Forest (KNF) and Manitou Experimental Forest (MEF) study units, reported as a mean (standard deviation) of 0.01 ha sampling unit. The locations are reported as site centroids using North American Datum 1983 Universal Transverse Mercator in Zone 12 North for KNF and Zone 13 North for MEF.

| Study Area | Northing | Easting | QMD (cm) | Max Tree Height (m) | Basal Area (m$^2$ ha$^{-1}$) | Trees ha$^{-1}$ | AGB * (Tons ha$^{-1}$) |
|---|---|---|---|---|---|---|---|
| KNF1 | 4,044,670 | 380,592 | 30.3 (14.8) | 15.9 (8.0) | 26.9 (22.0) | 300 (197) | 90.6 (51.1) |
| KNF2 | 4,044,484 | 380,496 | 31.2 (22.0) | 14.9 (9.6) | 21.2 (22.1) | 200 (186) | 80.7 (55.6) |
| KNF3 | 4,044,305 | 380,406 | 32.9 (14.6) | 22.2 (6.2) | 44.5 (29.2) | 626 (446) | 128.9 (54.7) |
| MEF1 | 4,330,850 | 490,190 | 21.7 (11.8) | 17.5 (6.6) | 24.8 (15.9) | 931 (806) | 90.2 (34.9) |
| MEF2 | 4,330,730 | 490,040 | 23.5 (11.3) | 17.1 (5.4) | 26.9 (17.4) | 701 (407) | 93.4 (35.1) |

* Aboveground biomass calculated using Jenkins et al. [21].

All trees > 1.37 m tall were stem-mapped from a grid of known survey locations at each study unit. The species, diameter at breast height (1.37 m; DBH), and height were recorded for each mapped tree. Stem mapping of the 60 m × 100 m study units (0.6 ha) was completed in May 2018 for KNF1, KNF2, and KNF3, then in July 2018 for MEF1 and MEF2. The stem maps were divided into 10 m × 10 m (0.01 ha; n = 60) sampling units. For each sampling unit, the above-ground biomass was estimated using Equation (1) from Jenkins et al. [21] as implemented in the Central Rockies variant [22] of the Forest Vegetation Simulator [23].

$$ABG = e^{(b_0 + b_1 \ln(DBH))} \tag{1}$$

where *AGB* is the total above-ground biomass (kg), *DBH* is in cm, and $b_0$ and $b_1$ are species-specific coefficients.

### 2.2. UAS Data Acquisition

UAS image data was collected using a DJI Phantom 4 Pro (Dá-Jiang Innovations Science and Technology Co. Ltd., Shenzhen, China) equipped with a 20-megapixel (5472 × 3648 pixels) metal oxide semiconductor (CMOS) red-green-blue (RGB) sensor, with a fixed 8.8 mm focal length. The aircraft recorded geolocation (x, y, and z) for each captured photo to a manufacturer-stated vertical accuracy of ±0.5 m and horizontal accuracy of ±1.5 m in World Geodetic System 1984. The camera was set to infinity focus for all image acquisitions, 5.6 aperture (F-stop), 1/500 s shutter speed, and ISO values of 100 to 200 depending on lighting conditions.

Flight planning and control used Altizure version 4.6.8.193 (Shenzhen, China) for Apple iOS to pre-program automated UAS flight paths at the desired altitude, forward and side photo overlap, and flight speed. All UAS surveys were flown between April and August 2019 within three hours of solar noon to maintain a minimum solar angle of 50° from the horizon. This study utilized a nadir camera angle, with 90% forward and side photo overlap. To evaluate how altitude impacts photogrammetric models of forest biomass, 40 UAS flights were planned (8 acquisitions per study area) at randomly chosen

altitudes ranging from 40 to 120 m above ground level. Flight speeds of 2, 3, and 4 m s$^{-1}$ were systematically assigned to each altitude. Flight boundaries were adaptively increased to maintain 10 flight lines and photo density (or the number of photos viewing the same location) at each desired altitude. This design resulted in flight boundaries varying between 80 m × 110 m and 161 m × 110 m at flight altitudes of 40 and 120 m, respectively. At the Kaibab study units, it was determined that the three lowest altitude (<45 m) UAS surveys could not be safely completed due to the proximity of the forest canopy, resulting in 37 total flights.

A total of ten Ground control points (GCP) were established at each site using high visibility ~0.2 m$^2$ markers that were located using a Trimble GeoXT (Trimble Inc., Sunnyvale, CA, USA) with SBAS real-time correction for each study unit with accuracies of <1 m. GCPs were placed to maximize visibility from the air with the UAS. Four points were set as close to each corner as possible, one along each long edge and the remaining four points distributed throughout the center. The GCPs were differentially corrected using Trimble Pathfinder Office post-processing software.

### 2.3. UAS Structure from Motion Point Cloud Generation Data Processing

Agisoft Metashape version 1.5.3 (Agisoft LLC, St. Petersburg, Russia) was used to generate 3D point clouds from an SfM photogrammetry algorithm. The imagery was processed following the procedures outlined in Tinkham and Swayze [24] to balance SfM forest crown reconstruction with data processing efficiency. The full suite of selected Agisoft Metashape settings for image dataset processing is available in Supplemental Table S1. Following point cloud generation, CloudCompare version 2.10.1 was used to visually inspect each point cloud to ensure complete dense cloud reconstruction. Agisoft Metashape processing reports were generated and checked to ensure similar processing errors across the point cloud models. Clouds with significant processing errors or incomplete reconstruction were reprocessed to ensure comparable accuracy and quality across the 37 surveys.

### 2.4. LiDAR Datasets

At the Manitou Experimental Forest sites, aerial LiDAR data was acquired by an Optech Pegasus HA500 operating at an altitude of 2000 m in August 2016. The MEF data was provided in North American Datum 1983 Lambert Conformal Conic with North American Vertical Datum 1988 at a nominal point density of 5.8 points m$^{-2}$ and vendor reported vertical accuracy of ±0.161 m at a 95% confidence level. Aerial LiDAR data for the Kaibab National Forest sites were acquired by an Optech Galaxy Prime operating at an altitude of 1800 m above ground level in the winter of 2019. The KNF data was provided in North American Datum 1983 Universal Transverse Mercator Zone 12 North with North American Vertical Datum 1988 at a nominal point density of 19.5 points m$^{-2}$ and a vendor reported vertical accuracy of ±0.326 m at a 95% confidence level. The time difference between the LiDAR acquisitions and the field inventory corresponds to average tree height growths of 0.25 m at MEF and 0.2 m at KNF, derived from prior site inventories. The LiDAR point clouds were cropped to the five study unit extents and used as a baseline for comparing the accuracy of the UAS modeled AGB.

### 2.5. Point Cloud Processing

All 37 SfM UAS and five aerial LiDAR point clouds were processed using the lidR package (version 3.2.3 [25]) in the R statistical programming language [26]. All datasets were transformed to North American Vertical Datum 1983 Universal Transverse Mercator in Zone 12 North for KNF and Zone 13 North for MEF. First, a stock cloth simulation filter was used to classify ground points, noise points were removed, and the point clouds were height normalized using the classified ground points with a k-nearest neighbor approach with inverse-distance weighting. The remaining points were classified as either ground or non-ground points. After filtering and classification, each point cloud was clipped to a 60 m × 100 m extent. The height normalized UAS point clouds were visually inspected for

horizontal alignment with their corresponding LiDAR point cloud in CloudCompare and all datasets were estimated to be within 0.50 m horizontal agreement.

The SfM points clouds were further processed to investigate how to point segmentation based on spectral indices impacts forest biomass modeling. To identify pure canopy and stem points, all filtered and classified point clouds at a study site were merged using lidR and loaded into CloudCompare. From the merged point cloud, 15 random samples of canopy and stem (each containing 20,000–60,000 points) were manually extracted to represent stem and canopy points. These classified stem and canopy point samples were processed in the lidR package [25], and the Normalized Green-Red Ratio (NGRR) was calculated for each point from the red and green image bands using Equation (2).

$$NGRR = (G_{band} - R_{band})/(G_{band} + R_{band}), \tag{2}$$

The stem and canopy point density plots of NGRR values for each study site (Figure 2) showed segregation of stem NGRR values at the 90th percentile and canopy NGRR values at the 10th percentile. Across study units, the stem 90th percentile of NGRR varied from $-0.0096$ to 0, and the canopy 10th percentile of NGRR varied from 0 to 0.0747. These 90th and 10th percentile segmentation of NGRR values were used to divide the SfM point clouds into three categories for each of the 37 acquisitions: Stem, Canopy, and Standard, where Standard is the non-NGRR classified point cloud.

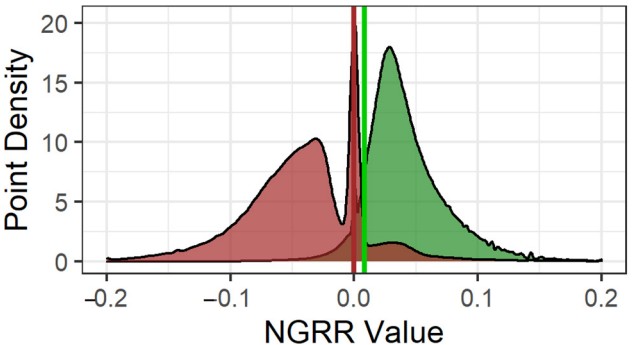

**Figure 2.** Example density plot of Canopy (green) and Stem (red) NGRR values sampled from SfM point clouds at the MEF1 study area, with overlapping distribution segments are brown. Red and green vertical lines represent the 90th and 10th percentile of the Stem and Canopy NGRR values, respectively.

*2.6. Forest Biomass Modeling*

To develop aboveground biomass models, the three SfM point clouds for each of the 37 UAS acquisitions and the five LiDAR point clouds were processed using the grid metrics utility in the lidR package. This function generated 35-point cloud distributional metrics for each 10 m × 10 m sampling units (Figure 1) used to derive biomass estimates from FVS. Metrics included maximum, average, standard deviation, skewness, kurtosis of heights, percent of heights above the mean and a 2 m threshold, percentiles (5 through 95 in steps of 5), and cumulative percentage of return in nine canopy layers [27].

The LiDAR point cloud distribution metrics were used to create baseline predictions of AGB using the Random Forest [28] and rfUtilities [29] packages of the R statistical programming language. The full random forest regression model [30] derived from 1000 trees from each dataset was used to predict AGB, where the 60 sampling units were randomly divided into training observations (80% or n = 48) and validation observations (20% or n = 12). To compare important point cloud predictor variables between the Standard and Standard + NGRR datasets, the Random Forest Model Selection tool [31] was used to identify each model's top five predictors. The same process for modeling AGB was repeated for each UAS data acquisition using the Standard SfM point cloud distribution metrics and

then again with the point cloud distribution metrics combined for the Standard, Stem, and Canopy datasets, referred to as Standard + NGRR hereafter.

### 2.7. Model Evaluation

The 74 SfM random forest models (37 Standard SfM and 37 Standard + NGRR SfM) were evaluated by relativizing them against their respective LiDAR random forest model by calculating the percent change in model performance using the Coefficient of Determination ($\Delta R^2$; Equation (3)), Root Mean Squared Error ($\Delta RMSE$; Equation (4)) and Mean Absolute Error ($\Delta MAE$; Equation (5)).

$$\Delta R^2 = (\text{SfM } R^2_{ij} - \text{LiDAR } R^2_i)/\text{LiDAR } R^2_i \times 100, \tag{3}$$

$$\Delta RMSE = (\text{SfM } RMSE_{ij} - \text{LiDAR } RMSE_i)/\text{LiDAR } RMSE_i \times 100, \tag{4}$$

$$\Delta MAE = (\text{SfM } MAE_{ij} - \text{LiDAR } MAE_i)/\text{LiDAR } MAE_i \times 100, \tag{5}$$

where i denotes an individual study site, and j signifies an individual UAS acquisition at that site. Positive values of $\Delta R^2$ indicate improvement of the UAS model over the LiDAR model. In contrast, negative values of $\Delta RMSE$ and $\Delta MAE$ indicate reductions in the UAS model compared to the LiDAR model. To standardize the effect of altitude on model performance across the five sites, altitude was evaluated as a ratio ($A{:}L_H$; Equation (6)) of altitude (A) compared to Lorey's Mean Height ($L_H$; Equation (7) [32]).

$$A{:}L_H = \text{Altitude (m)}/L_H \text{ (m)}, \tag{6}$$

$$L_H = \sum (g \times h)/\sum g, \tag{7}$$

where g is a tree's basal area ($m^2$), and h is a tree's height (m), meaning $L_H$ can be interpreted as the weighted height of the forest where stands with more regeneration will have a value less than the mean.

Linear mixed-effects regression evaluated the relationship between Standard UAS random forest model performance metrics with UAS data acquisition flight altitude and speed. In this analysis, $\Delta R^2$, $\Delta RMSE$, and $\Delta MAE$ were predicted from the 37 combinations of UAS acquisition altitude and speed were treated as fixed effects, while the five study sites were treated as a random effect. Additional covariates of stand-level forest structure (see Table 1) and AGB were also evaluated for their influence on model performance. While testing for interactions, a stepwise procedure was used to identify the best subset of explanatory factors that minimized the Akaike Information Criterion (AIC). All regressions were performed using the lme4 package [33] of the R statistical programming language. A second set of models was fit using $A{:}L_H$ in place of altitude above ground. The distributions of $\Delta R^2$, $\Delta RMSE$, and $\Delta MAE$ were compared between the Standard and Standard + NGRR models using paired Wilcoxon signed-rank tests to evaluate if the two sets of model predictors resulted in differences in model performance.

Finally, those point cloud distribution metrics listed above that were important in each random forest model were pooled and evaluated for their frequency across models. This analysis was extended to contrast how important distribution metrics from the Standard SfM models differed in the Standard + NGRR SfM models.

## 3. Results

### 3.1. LiDAR AGB Model Performance

The five-baseline aerial LiDAR random forest model results of AGB varied across the study sites, with $R^2$ averaging 0.502 (range 0.371–0.569). The mean LiDAR $R^2$ values for the two MEF study sites were, on average, marginally higher (0.068) than those for the three KNF study sites. Compared to the MEF sites, the KNF study sites had nearly twice as much variation in the 0.01 ha sampling units for AGB, basal area ha$^{-1}$, and maximum tree heights (Table 1). Similar contrasts in model performance were found in RMSE, which ranged from

27.2 to 58.0 tons ha$^{-1}$, and MAE, which ranged from 20.5 to 44.2 tons per ha$^{-1}$. Variation in these metrics across the five study sites followed the same trend as the R$^2$ values.

### 3.2. UAS AGB Model Performance

Differences in AGB model performance between the baseline LiDAR and Standard UAS SfM varied across flight altitudes (Figure 3). The low altitude UAS acquisitions failed to adequately reconstruct the vegetation's vertical profile, resulting in worse results (average ΔR$^2$ = −36.9%). For acquisitions at higher altitudes, the average ΔR$^2$ was 7.7%. Generally speaking, we found that model performance improved with increased altitude (Table 2 and Figure 4), where average ΔR$^2$ ranging from a 25% reduction in prediction performance at the lowest altitudes to nearly a 20% improvement over the LiDAR models at the highest altitudes (Table 2 and Figure 4). Similar improvements with increased altitude were seen for Standard UAS SfM models of AGB for ΔRMSE and ΔMAE (Figure 3). However, no effect of flight speed on ΔR$^2$ was identified.

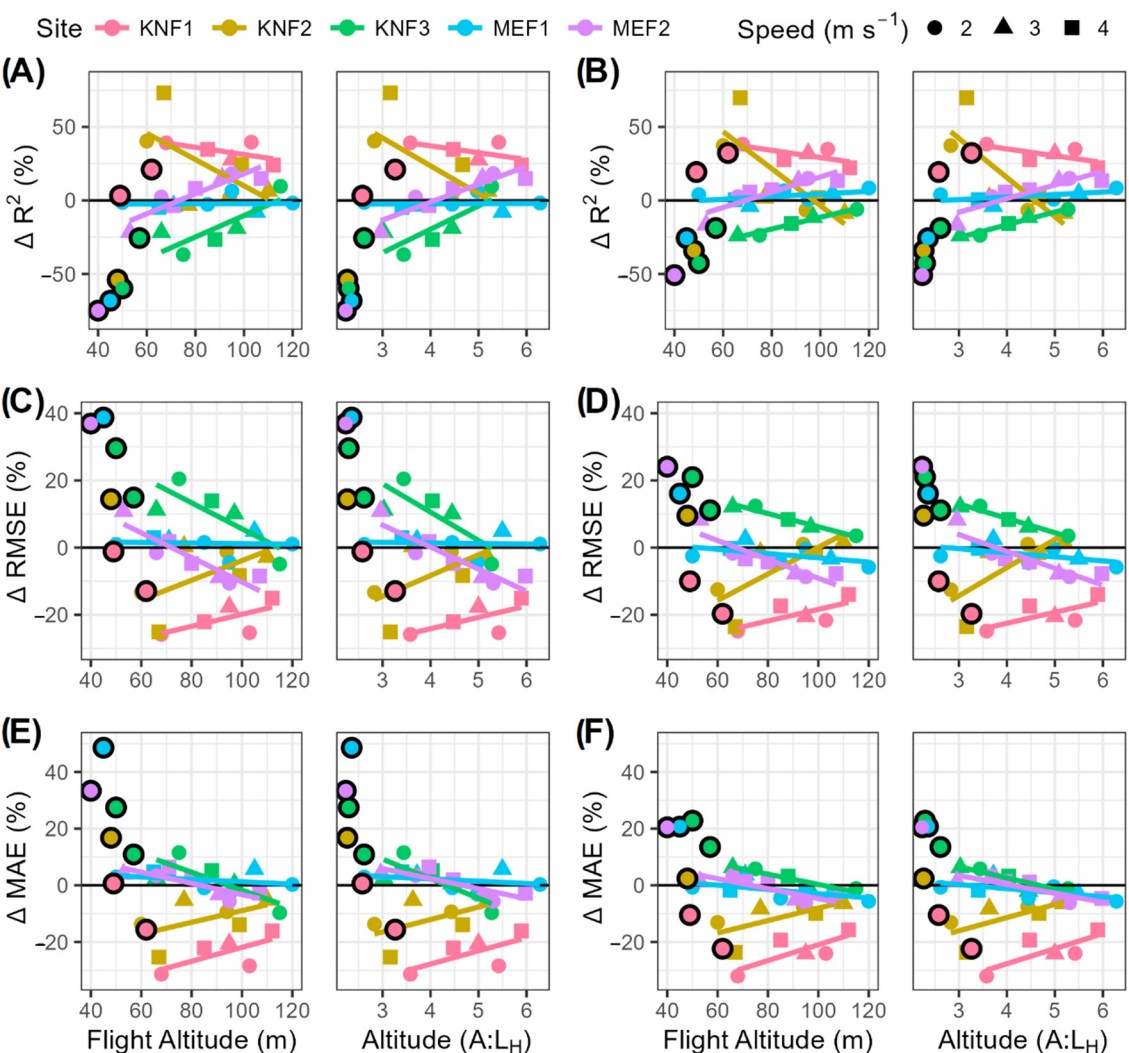

**Figure 3.** Comparison of Standard (panels (**A,C,E**)) and Standard + NGRR (panels (**B,D,F**)) AGB model performance metrics relativized to LiDAR, including percent ΔR$^2$ (panels (**A,B**)), ΔRMSE (**C,D**), and ΔMAE (**E,F**). The panel is split to show the influence of altitude above ground (left) and the ratio A:L$_H$ (right). Points in black circles represent acquisitions that failed to reconstruct the forest canopy fully and were therefore excluded from the best it lines.

**Table 2.** Linear mixed-effects model of flight altitude and speed influence the change in aboveground biomass model $R^2$ ($\Delta R^2$). A total of four models were created for combinations of the 37 Standard or 37 Standard + NGRR predictions with altitude or relative altitude as predictors; the five study sites were treated as a random effect.

| Parameter | Coefficient | SE | *p*-Value | Coefficient | SE | *p*-Value |
|---|---|---|---|---|---|---|
| | *Standard Parameters* | | | *Standard + NGRR Parameters* | | |
| Intercept | −56.358 | 21.812 | 0.0143 | −28.447 | 17.689 | 0.1175 |
| Altitude (m) | 0.596 | 0.180 | 0.0024 | 0.2843 | 0.143 | 0.0556 |
| Speed (m s$^{-1}$) | 3.081 | 5.076 | 0.5483 | 3.184 | 4.037 | 0.4364 |
| | *Standard Parameters* | | | *Standard + NGRR Parameters* | | |
| Intercept | −56.862 | 21.390 | 0.0120 | −29.996 | 17.328 | 0.0930 |
| Relative Altitude (A:L$_H$) | 11.933 | 3.471 | 0.0017 | 6.013 | 2.760 | 0.0372 |
| Speed (m s$^{-1}$) | 2.988 | 5.046 | 0.5581 | 3.144 | 4.012 | 0.4393 |

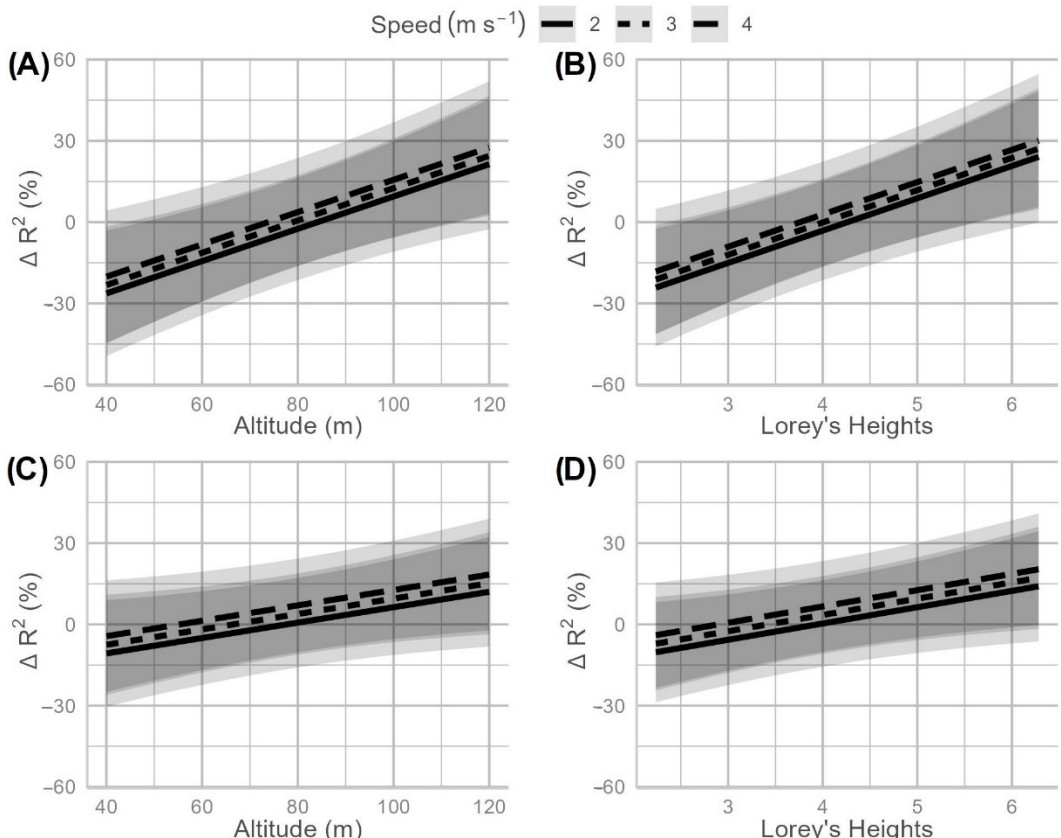

**Figure 4.** Linear mixed-effects model results for the 37 Standard SfM AGB models (panels (**A**,**B**)) and the 37 Standard + NGRR AGB models (panels (**C**,**D**)).

The altitude at which Standard UAS SfM models began outperforming LiDAR models varied from ~80 to 100 m (Figure 3). Standardizing flight altitude as a ratio of altitude divided by Lorey's Height narrows the range of altitudes at which standard UAS SfM models outperform the LiDAR models to 4–4.5 times the site's Lorey's Height. Just as flight altitude, using Lorey's Height to relativize altitude significantly explains the variation in $\Delta R^2$ (Table 2). The linear mixed-effects model indicates that for every Lorey's Height acquisition altitude increases, there is an ~12% improvement in $\Delta R^2$, meaning that at four times Lorey's Height, UAS SfM modeling of AGB typically exceeds the performance of aerial LiDAR models in ponderosa pine-dominated systems (Figure 4).

Using the Standard + NGRR datasets for AGB modeling provided a 3.7% average increase in $\Delta R^2$ across all flights compared to the Standard dataset modeling (Figure 3). When tested with a paired Wilcoxon signed-rank test, this increase was not significant

(*p*-value = 0.0750). However, there was a slight improvement in ΔRMSE (*p*-value = 0.0567) by 2.2% for the Standard + NGRR models and a significant (*p*-value = 0.0092) decrease in ΔMAE by 2.8% compared to models only using the Standard SfM parameters. The linear mixed-effects model of ΔR² for the Standard + NGRR predictions followed similar trends to the Standard predictions (Figure 4) but with a slightly shallower slope. However, the Standard + NGRR predictions had an intercept much closer to zero (only marginally departing from zero when predicting based on relative altitude) than the Standard predictions (Table 2).

### 3.3. Comparison of Point Cloud Structure

Clear trends in the impact of altitude on UAS SfM modeling of AGB are present in the results. Comparisons of point cloud distributions from UAS SfM from the lowest altitudes flown differ in reconstructing the upper canopy within each stand (Figure 5). Conversely, UAS SfM point cloud distributions from higher altitude flights exhibited trends similar to the LiDAR datasets or provided slightly greater point density in the mid- and lower-canopy (Figure 5).

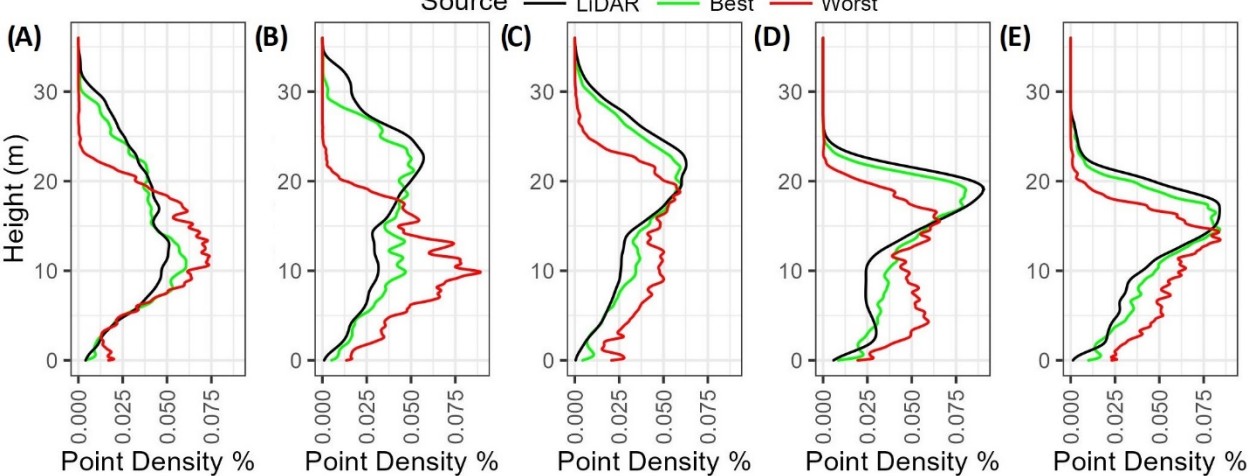

**Figure 5.** Relative density as a function of height above ground for the models providing the best and worst model ΔR² from the Standard SfM AGB datasets compared to the LiDAR at each site (from left to right (**A**) KNF1, (**B**) KNF2, (**C**) KNF3, (**D**) MEF1, and (**E**) MEF2).

For the Standard SfM models of AGB, the five most important variables fluctuated across models, but five variables stood out from the rest and showed up in at least 34% of these models (Figure 6A). Across all flight altitudes and sites, the percentage of points above 2 m height and the average height of points above ground showed up in 71.4 and 65.7% of all models, respectively. The remaining three frequently selected variables were in 34 to 40% of models, with their importance metric levels fluctuating. These five metrics accounted for 46.5% of the top five most important metrics selected in all the Standard SfM models.

Of the top five variables from the Standard SfM metrics, the top three remained unchanged even after including the NGRR segmented point cloud metrics (Figure 6B). The other higher tier metrics also came from the list of Standard metrics. However, the five most important metrics now only accounted for 37.8% of all selected metrics when the Standard and NGRR metrics were considered. This reduction was due to 16.8 and 10.2% of all selected variables being chosen from the Stem and Canopy distributional metrics, respectively. Of these NGRR metrics, only the maximum height and average height of Stem points and maximum height of Canopy points were important in more than 15% of models (Figure 6B).

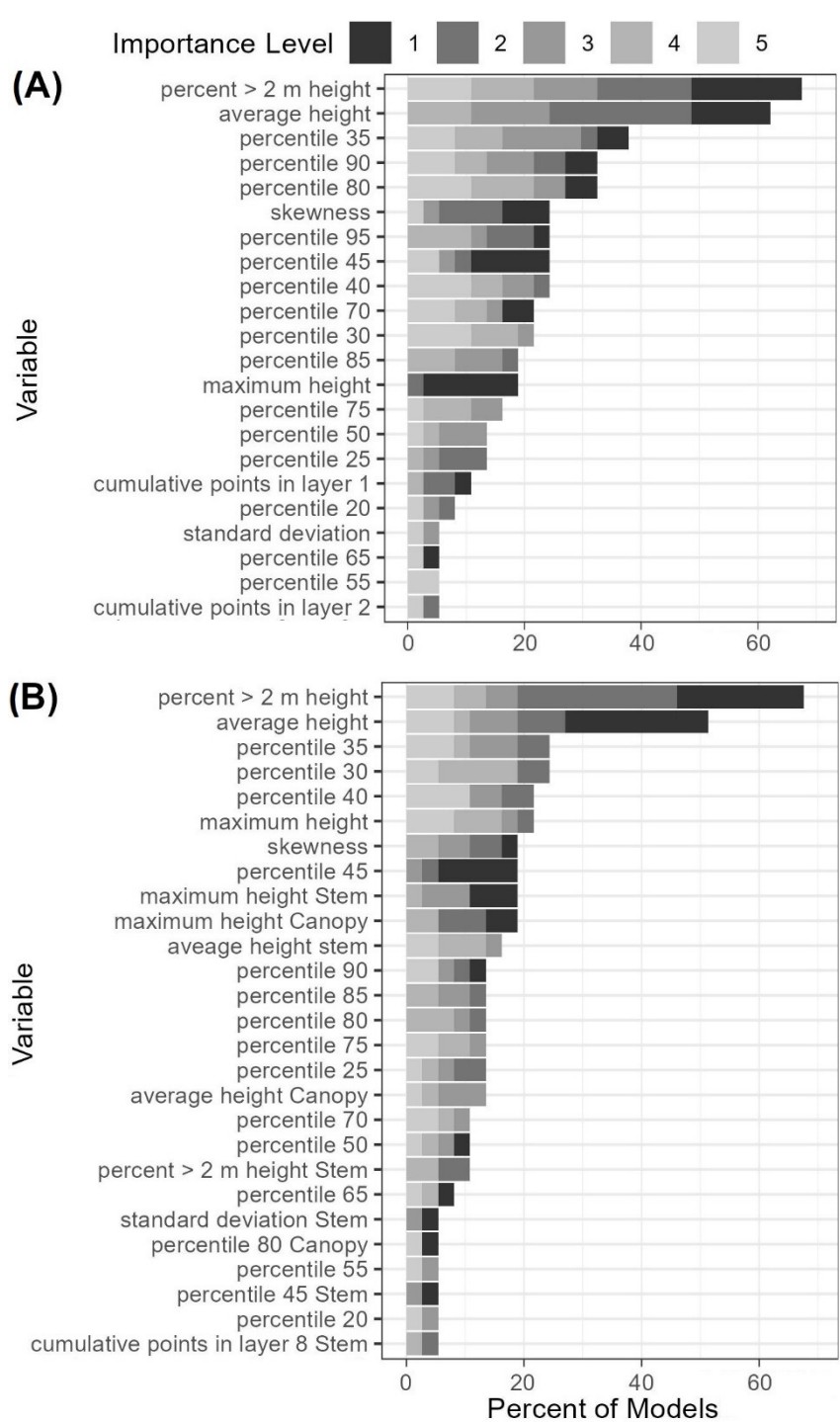

**Figure 6.** Comparison of the five most important Random Forest variables for the Standard (**A**) and Standard + NGRR (**B**) datasets, values are sorted based on the percentage of the 37 models they appeared in.

## 4. Discussion

### 4.1. AGB Model Performance

This study evaluated the impact of flight altitude and speed on UAS SfM plot-based modeling of aboveground biomass, with UAS modeling at higher acquisition altitudes outperforming LiDAR. Across all sites, UAS acquisitions above 80 m led to $\Delta R^2$ of 9.6% and reduced RMSE and MAE by 5.7 and 7.1%, respectively (Figure 3). While the authors could not identify other studies with this relationship for UAS biomass modeling, inference

might be drawn from studies looking at relationships between UAS SfM reconstruction quality and flight altitude. Fraser and Congalton [13] tested the effect of flight altitude on image alignment and found that flying at their highest tested altitude provided the best results. Other studies have not directly seen an influence of altitude on SfM vegetation reconstruction but still concluded that flying higher provided the benefit of greater acquisition extents [12]. This literature connects the slight decrease in image resolution at higher altitudes with improved image matching by reducing the influence of vegetation movement in the wind [34]. There is reason to believe this improved image matching represents vertical vegetation distributions better and improves AGB modeling at the plot level. Our results point to UAS plot-based modeling of AGB being a viable alternative in forest systems where the cost of aerial LiDAR might be prohibitive. However, the time lag between the LiDAR and stem map data for the two MEF sites may have confounded the exact magnitude of UAS model improvement.

Part of the improved UAS SfM model performance with increased altitude can be attributed to differences in the photogrammetric reconstruction of the upper canopy for the lowest altitudes tested at each site. The dependence of UAS SfM reconstruction on altitude is attributed to the proximity of treetops to the sensor. When the vegetation is too close to the sensor, there is insufficient photo overlap at the top of the tree compared to the programmed 90% forward and 90% side overlaps at the ground surface. Although lower altitude UAS acquisitions can provide greater resolution, studies that require close proximity acquisition of forest canopies should consider increasing the photo overlap to compensate for the depth of vegetation. Additionally, atmospheric stability will decrease the closer to forest canopy a UAS operates, potentially leading to image blurring and impacting image alignment [35]. Our results indicate that consistent point cloud generation is achieved for flights above four times Lorey's Height. Therefore, as the height of vegetation increases, the flight altitude needed to reconstruct forest canopies should also increase. However, it is likely that improvements in UAS model performance with increased altitude will plateau. At some point, increases in flight altitude beyond this point will decrease image resolution resulting in reduced performance in modeling forest attributes such as AGB. Further research is needed to understand the transference of our four times Lorey's Height threshold for AGB modeling in shorter and tall forest systems.

Relativizing altitude as a ratio of Lorey's Height (A:$L_H$) for each flight provided better consistency for interpreting the relationship between UAS flight altitude and vegetation height. All flights above four times a site's Lorey's Height improved model performance, with an average increase of 9.6% for $R^2$ and decreases of 5.7 and 7.1% for RMSE and MAE, respectively (Figure 3). Currently, it is difficult to evaluate the effects of altitude in different vegetation types, as most studies do not report vegetation height or its relationship to flight altitude. Standardizing the reporting of UAS acquisition and vegetation structure parameters is necessary within the UAS literature to improve the cross-study synthesis of results.

Our results did not find a statistically significant effect of flight speed on resultant AGB models (Table 2), which could be attributed to the narrow range of relatively slow flight speeds (2–4 m s$^{-1}$) evaluated. These findings differ from O'Connor et al. [36], which found that increased flight speed causes image blurring and location errors, resulting in image alignment errors. While there is reason to believe this should propagate through the modeling of forest attributes such as AGB, it did not manifest in the flight speeds tested in this study. Although not significant in this study, flight speed should remain an important consideration in planning UAS-based forest remote sensing. The effects of flight speed on modeling forest attributes such as AGB need to be evaluated across a broader range of speeds and cross-compared between sensors (rolling vs. global shutter) and UAS platform types (multi-rotor vs. fixed-wing [37]).

The inclusion of the NGRR point cloud distribution metrics improved the prediction of AGB in terms of variance explained ($\Delta R^2$) and precision ($\Delta$RMSE) over the Standard point cloud predictions. When relativized, the magnitude of $\Delta R^2$ improvement of NGRR

point clouds over the Standard models was only ~2%, but the ΔRMSE was improved by ~12%. When NGRR metrics were included, they accounted for ~29% of important random forest predictors, with ~19% coming from the Stem point cloud metrics. Comparison of the Standard and Standard + NGRR model performance metrics revealed no clear trend across the UAS acquisition parameters. Despite using only RGB spectral information, the improved model performance suggests that spectral segmentation of photogrammetric point clouds may be a powerful tool for improving models of forest structural attributes. Other studies share our results, which found that including spectral indices from image orthomosaics as predictors of forest structure and biomass significantly improved model performance [16]. There is reason to believe more advanced segmentation and characterization of SfM points beyond indices available from RGB imagery could further improve the modeling of forest biomass done in this study. Including a greater range of spectral data from more powerful multispectral sensors may provide better discernment between vegetation structural components within SfM point clouds.

### 4.2. Implications for Forest Management

This study highlights the potential of UAS SfM plot-based AGB modeling to match or exceed aerial LiDAR modeling. However, it also emphasizes the importance of flight altitude and speed selection on UAS model reliability. UAS based-data collection approaches are increasingly used to characterize forest structure and biomass across many ecosystems and forest management objectives [38–40]. The growing focus on UAS technology is attributed to its high temporal and spatial resolution at relatively low operational costs compared to similar datasets from aerial LiDAR. The reduced price of UAS remote sensing could make AGB monitoring a viable mechanism for managing low productivity forest and woodlands and other ecosystems where more common aerial LiDAR approaches are cost-prohibitive.

The potential of flying at heights 4–6 times a stand's Lorey's Height should be expected to result in faster data collection and processing times and reduced data storage demands [19]. The wider image footprints obtained at higher altitudes in this study provided more efficient flight times (~2 min ha$^{-1}$), with 1 ha being acquired in 2.5 min and 5 min on average for altitudes of 100 m and 60 m, respectively. Additionally, since moderate increases in UAS flight speed did not detrimentally impact model performance (Table 2), flying at 4 m s$^{-1}$ could further improve the rate of data acquisition. These results indicate that rapid and reliable plot-based AGB modeling from UAS SfM is possible.

This study's plot-based UAS modeling strategy effectively describes the vertical distributions of forest attributes, potentially lending the data to modeling other forest attributes. Following developments in the aerial LiDAR literature, extending UAS modeling to other forest attributes such as basal area, volume, and successional stages should find similar success to this study. While this strategy successfully described plot and stand-level AGB, further exploration of the potential for UAS-based systems to characterize forest structure at both tree and landscape scales is needed. The ultra-high resolution of UAS data products and potential to fuse spectral and structural characteristics should enable improved individual tree characterizations. Early testing of single tree extraction methods from UAS SfM data has successfully identified >90% of trees [41,42]. These individual tree characterization strategies perform best when canopy cover is < 50% [43]. Additionally, UAS has the potential to serve as a sampling tool, potentially providing vast amounts of data to train coarser landscape-scale satellite-based models of woodland, savannas, and forest biomass. Recent research has demonstrated techniques for scaling UAS observations to describe biomass at greater extents than UAS were capable of characterizing [44].

### 5. Conclusions

This study demonstrates the potential of plot-based UAS photogrammetry for modeling aboveground biomass in low productivity forests and woodlands. The UAS modeling performance was optimized when flying at altitudes greater than four times a forest's

Lorey's Height, resulting in an average 7.8% improvement in $R^2$ over aerial LiDAR. Additionally, segmenting the SfM point cloud based on image spectral signatures tied to individual points to describe the distribution of Stem and Canopy points provided further improvements to the AGB modeling. This study highlights the role of UAS acquisition parameters on plot-based forest biomass modeling while also showing strong potential for UAS-based forest monitoring at increased temporal frequencies than have been feasible from aerial LiDAR. Operationalizing UAS monitoring in low productivity forests and woodlands could provide access to alternative management funding strategies but will require full vetting of the method through the monitoring, reporting, and verification process required by carbon markets.

**Supplementary Materials:** The following are available online at https://www.mdpi.com/article/10.3390/rs14091989/s1, Table S1: Agisoft Metashape processing parameters for SfM photogrammetry forest reconstruction.

**Author Contributions:** Conceptualization, N.C.S., W.T.T. and M.B.C.; methodology, N.C.S. and W.T.T.; data processing, N.C.S.; formal analysis, N.C.S. and W.T.T.; resources, W.T.T. and C.M.H.; writing—original draft preparation, N.C.S., W.T.T. and M.B.C.; writing—review and editing, C.M.H., J.C.V. and A.T.H.; supervision, W.T.T. All authors have read and agreed to the published version of the manuscript.

**Funding:** This research was funded by the United States Department of Agriculture McIntire-Stennis Capacity Grant (COL00511).

**Data Availability Statement:** The data presented in this study are available on request from the corresponding author.

**Acknowledgments:** The authors would like to thank Mike Battaglia, Wayne Shepperd, and Lance Asherin for establishing and maintaining the stem-mapped study sites. Additionally, we would like the thank Alex Weissman, Taylor Richmond, Alexis Conley, Alexa Binkley, Adam Langemeier, Brandon Hoem, Jillian LaRoe, and Steven Filippelli.

**Conflicts of Interest:** The authors declare no conflict of interest.

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
