# Peer review of "Influence of UAS Flight Altitude and Speed on Aboveground Biomass Prediction"

_remotesensing, doi:10.3390/rs14091989_

Round 1
Reviewer 1 Report
In this article, Swayze et al. analyzed different flight parameters (altitude, speed) of UAVs in order to assess how those parameters affect forest biomass evaluation, in contrast to LiDAR surveys using airplanes. They explored one software doing SfM to extract some point cloud and compared it to the LiDAR point cloud.The conclusions of this study could have some importance for the estimation of forest biomass in agroforestry and, by ricochet, on the carbon footprint and forest productivity. One improvement that the authors propose is the inclusion of the NGRR index in the computation of the point cloud. Unfortunately, there are some important results and information missing from this manuscript. For example, while the title emphasis AGB calculations, no results are presented about this estimation for the various methods the authors tried, and the actual calculations are not stated. One major problem in this study is the use of LiDAR point generated at very different dates (August 2014, Winter 2019, lines 190 and 191) which are being compared to SfM point cloud from 2018 (line 141). Overall, the subject is interesting, but the manuscript lack some important information to the reader in order to really compare UAV SfM vs aerial LiDAR estimation for biomass estimation in forest.
General comments:
1- In Figure 1 legend, indicate the size of each subplot.
2- What is the importance of Figure 2 in this manuscript?
3- A histogram of the distribution of AGB in relation to the different flight parameters need to be included in this manuscript.
4-Figure 5. What is the calculation (formula, threshold and citation) used to classify the Best and Worst relative density for the point cloud distribution presented?
5- It would be critical for this manuscript to include the equation and calculation of the AGB using the different points clouds created vs LiDAR in order to really compare the different methods.
6- Could the authors comment on why they did not try some UAV mounted LiDAR instead of using completely different technology and methodology ?
7-In this study, the authors only compare one software for the SfM (Agisoft Methashape). Since the software seems critical to the results of this study (especially since it seems unable to reconstruct scenes at slow speed and low altitude - Figure 3), could the author discuss if they have the same results using other software (Pix4D, WebODM, etc…)?
8- The authors chose to try the NGRR index to help in generating the SfM model, do other vegetation index achieve the same results (e.g. NDVI)?
9-At line 312, please refrain from saying “marginally significant” when it is not.
10- Seems like the importance of variables (Figure 6) are quite similar. Maybe a table would be more appropriate to distinguish the difference between the importance of each available for the two datasets.
11-In a forest, there are different tree species having different geometry, leave shapes, etc. Did the authors found a correlation between some species and better correlation between the point clouds?
9-Reference 31 (line 238) seems wrong since it does not refer to a RandomForest software or package.
10-There is some major difference between the LiDAR acquisition date and drone survey. For example, some data were acquired in winter (no leaves?) vs summer. This should be discussed with great details in the manuscript since it seems very unlikely that it does not affect the results.
11-Is there a threshold in the biomass (g, or LH) where the estimation by both methods are either very dissimilar, very similar?
12-Please revise the title if AGB is not compared in the manuscript. It should be the "Influence of UAS Flight altitude and speed on point cloud density".
Reviewer 2 Report
This is a very well-written paper and covers a lot of ground in detail. The methodology appears to be sound and the paper is well illustrated and documented. The discussion is also useful as is the reference to additional future research. There are a few minor slightly clunky sentences that could be edited with a final read over. The only things I might have expected to see are a few more study site photos from the ground level, a study area map, some illustrations relating to the GPS GCPs, maybe some images of the kit and in addition perhaps a flowchart of the methodology - although this of course may make the paper too long! Overall, only minor edits needed. An interesting paper and the coverage of the technical aspects of a drone survey application are a useful addition to the literature. But a pleasure to read.
Reviewer 3 Report
remotesensing-1628224-peer-review-v1
The manuscript “Influence of UAS Flight Altitude and Speed on Aboveground Biomass Prediction” addresses an interesting and up-to-date subject, which adhere to Remote Sensing journal policies.
The manuscript tackles an interesting topic, and presents a good remote sensing application, but certain methodological aspects are questionable. While the research is well conceived and written, the manuscript needs improvements and additional considerations from the authors:
- It is unclear the datum used in your data acquisition
- The LiDAR data used is not clearly explained. It was aerial LiDAR from 2014 but what kind of sensor, what was the vertical accuracy of the captured data (very important in your future analysis vs UAV)
- Also, the time difference between LiDAR data (2014) and UAV mission flights (2019) is not ideal
- At R151 I would drop the manufacturer stated accuracies, as they are much lower (in good conditions ±5m planimetric and even lower altimetric). That is why GCPs & CPs are so important
- The GCPs in any SfM photogrammetry process is very important. While the average number of CGPs / plot sample is adequate (10), the measurement/instrumentation for the obtained coordinates is inadequate. Trimble GeoXT is a handheld GPS with low accuracy (~ 1 m planimetric and even lower on elevation). GCPs must be measured using survey-grade instrumentation, such as a GNSS system, total station etc.
The scope, analysis & discussion is good, but it is unfortunate that the data acquisition is not ideal. The datasets compared could have been affected by these issues and some results misleading. Maybe the authors can further elaborate/improve.
Reviewer 4 Report
The paper entitled “Influence of UAS Flight Altitude and Speed on Aboveground 2 Biomass Prediction” reflects the development of applied research, the topic is interesting and the manuscript has an approach innovative. The manuscript is well structured, methods, results and discussion are appropriated. Yet, there are some issues that can be improved (see comments). Thus, it is recommended minor changes.
Comments
1) Line 112-113 and 126 – Please consider adding the central coordinates of study sites.
2) The symbol for ton is t.
3) Figure 2 – Red and green vertical lines represent the 90th and 10th percentile of the Stem 222 and Canopy NGRR values, respectively
4) R statistical program. The reference is missing
5) Line 254 – “Lorey’s Mean Height”. Please consider adding a reference.
6) Line 278 – max or maximum?
7) Line 311 – Wilcoxon test should be included in methods section.
8) Check for subscript and superscript mismatch along the text.
Round 2
Reviewer 3 Report
The revised manuscript demonstrates the author’s commitment in improving the overall paper, thus obtaining a cohesive and interesting article.
Also, the author's response to the reviewer's concerns were addressed satisfactorily. While the GCPs used in the photogrammetry process would not be suitable in monitoring or precision mapping, for a foresty application it is fine.
In my opinion, the manuscript can be considered for publication.
Author Response
We thank the reviewer for their constructive feedback and for providing insight on the best use case for ground control in future research efforts.